# Heme Degradation in Pathophysiology of and Countermeasures to Inflammation-Associated Disease

**DOI:** 10.3390/ijms21249698

**Published:** 2020-12-18

**Authors:** Donald David Haines, Arpad Tosaki

**Affiliations:** 1Advanced Biotherapeutics, London W2 1EB, UK; ddhaines2002@yahoo.com; 2Department of Pharmacology, Faculty of Pharmacy, University of Debrecen, 4032 Debrecen, Hungary

**Keywords:** heme degradation, heme oxygenase, diseases, therapies

## Abstract

The class of tetrapyrrol “coordination complexes” called hemes are prosthetic group components of metalloproteins including hemoglobin, which provide functionality to these physiologically essential macromolecules by reversibly binding diatomic gasses, notably O_2_, which complexes to ferrous (reduced/Fe(II)) iron within the heme porphyrin ring of hemoglobin in a pH- and PCO_2_-dependent manner—thus allowing their transport and delivery to anatomic sites of their function. Here, pathologies associated with aberrant heme degradation are explored in the context of their underlying mechanisms and emerging medical countermeasures developed using heme oxygenase (HO), its major degradative enzyme and bioactive metabolites produced by HO activity. Tissue deposits of heme accumulate as a result of the removal of senescent or damaged erythrocytes from circulation by splenic macrophages, which destroy the cells and internal proteins, including hemoglobin, leaving free heme to accumulate, posing a significant toxicogenic challenge. In humans, HO uses NADPH as a reducing agent, along with molecular oxygen, to degrade heme into carbon monoxide (CO), free ferrous iron (FeII), which is sequestered by ferritin protein, and biliverdin, subsequently metabolized to bilirubin, a potent inhibitor of oxidative stress-mediated tissue damage. CO acts as a cellular messenger and augments vasodilation. Nevertheless, disease- or trauma-associated oxidative stressors sufficiently intense to overwhelm HO may trigger or exacerbate a wide range of diseases, including cardiovascular and neurologic syndromes. Here, strategies are described for counteracting the effects of aberrant heme degradation, with a particular focus on “bioflavonoids” as HO inducers, shown to cause amelioration of severe inflammatory diseases.

## 1. Heme Structure, Function and Chemistry

Heme molecules are prosthetic coordination complexes produced in the liver and bone marrow, which serve as an oxygen-binding component for several classes of hemoproteins, a family of metalloproteins essential for tissue oxygenation through the circulation. The presence of a heme complex in hemoglobin is responsible for the characteristic red pigmentation of blood, since under normal physiological conditions, this protein is intracellular, found within erythrocyte (red blood cell; RBC) cytoplasms, with its release free into the plasma, typically a feature of pathological processes such as hemolysis occurring in sickle cell disease. This quality is reflected by the derivation of the name for the molecule: aqua haima, the term in Greek for “blood” [1]. As shown in Figure 1, heme structure prominently includes a porphyrin ring bound as a tetradentate ligand to a central iron atom, with either one or two axial ligands [2]. Other than hemoglobin, these complexes and related substituted porphines (porphyrins) also constitute the major functional prosthetic group in biologically important molecules such as myoglobins, endothelial nitric oxide synthase, catalases, cytochromes, chlorophyll and heme peroxidase [3]. The iron atom within the heme component of hemoproteins within the RBC cytoplasm binds reversibly to diatomic gasses diffusing into the cells from plasma, including but not limited to O_2_, causing conformational changes in the intracellular carrier protein which favor the particular biological function for which the protein is essential. Iron within heme groups also functions as a source of electrons for heme-mediated redox and electron transfer reactions and transport or detection of diatomic gases [4]. The binding selectivity of O_2_ and other ligands to iron is dependent on its oxidation state, with the ferrous form Fe(II) complexing with cognate ligands different from Fe(III) [4]. Heme-bound Fe(II) also provides a source of electrons for these processes, with the associated porphyrin group contributing to a reduction in the tissue toxicity of singlet/radical electrons through delocalization effects by the conjugated porphyrin ring [4]. These properties allow hemoproteins to participate in a diverse range of essential physiologic processes, with hemoglobin-mediated tissue oxygenation being the most extensively characterized at the time of this writing [5]. The highly diverse functionality of hemoproteins is due to structural features, notably the influence of specific amino acids in close proximity to the heme group that mediate conformational changes in the hemoprotein favoring pH-dependent O_2_ attachment to the iron through a mechanism called the Bohr Effect [5]. The relatively high pH and low CO_2_ conditions in lung microvasculature promotes O_2_ attachment to heme iron, and when this ratio is reversed in the tissues through CO_2_-associated pH decrease, O_2_ dissociates from the iron and diffuses to cells where it is needed [6].

## 2. Tissue Deposition and Degradation of Heme

The hemoprotein family are defined by the presence of heme within their structures and include endothelial nitric oxide synthase, catalases, cytochromes, heme peroxidase and myoglobin. Each of these macromolecules mediates physiological processes vital to the health of an organism. Within vertebrate systems, a significant amount of heme is localized in circulating erythrocytes, complexed with hemoglobin, which releases hemin when iron in its ferrous (Fe(II)) form is oxidized to a ferric (Fe(III)) state [8]. Moreover, as these cells become aged or damaged, they are removed from circulation by splenic macrophages in a process that results in further release of heme into the plasma and interstitium to pose a toxicogenic challenge through its tissue accumulation, where it may act as an oxidative stressor [9]. The lipophilicity of heme, which allows its intercalation into lipid bilayers, engenders both physical and chemical disruption of organelles and cells, compromising their integrity and adversely altering cytoskeletal structure. This results in endothelial cell injury and increased risk of vascular inflammatory disorders, with accompanying increased expression of intracellular adhesion molecules. Each of these effects may contribute to a wide range of debilitating chronic illnesses, including pathologies of the nervous system, heart, lungs, kidneys and liver [9]. The major adaptive countermeasure that has evolved to meet the challenge of this hazard is clearance of the molecule through its degradation by heme oxygenase (HO), a 32 kDa heat shock protein (hsp32) composed of 288 amino acid residues, found in the plasma membrane, nucleus and mitochondria of cells, with particularly high levels in the endoplasmic reticulum [10]. Heme is the physiologic substrate for HO, which uses NADPH, molecular oxygen and cytochrome p450 reductase to degrade heme into biliverdin, which is subsequently converted by the enzyme biliverdin reductase to bilirubin, carbon monoxide and ferrous iron Fe(II) as shown in Figure 1 [7]. This reaction occurs both in intra- and extracellular physiologic sites, thus providing cytoprotection more comprehensively than (for instance) dietary antioxidants, which act mostly in interstitial spaces [11]. Failure of endogenous hemoglobin and heme degradation mechanisms to effectively clear these molecules from tissue spaces may result in severely pathological consequences. Below, a schematic summary of the various processes contributing to such outcomes is provided with focus therein given to rigor of definitions for each of the participating cellular and macromolecular components shown in sequence of progression known to occur in pathogenesis of the model disease. This diagram shows the major consequences of degradative process failure: a schematic summary, using a model of sickling disease in human subjects. As shown in Figure 2, the major pathomechanisms of sickle cell anemia provide a representative model of cellular and molecular processes leading from the release of unreacted hemoglobin and heme into the plasma. For this particular disease, the progression of events shown below illustrates how the release of heme and hemoglobin into the plasma adversely affects vascular tissue.

## 3. Representative Aberrant Hemoglobin and Heme Degradation Pathway

The “line flow” blelow shows the major pathogenic events of sickle cell disease leading to abnormal tissue function and cell death.

Erythrocyte (RBC) sickling deformity >> hemolysis >> free hemoglobin and heme >> heme iron >> plasma components >> reactive oxygen species (ROS) >> exacerbated RBC destruction >> free hemoglobin and heme >> activated leukocytes >> inflammatory cytokines >> vascular endothelium >> NF-kB + ROS + endothelial cell adhesion molecule (ECAM) + vascular cell adhesion molecule-1 (VCAM-1), intracellular cell adhesion molecule-1 (ICAM-1) + P-selectin >> increased RBC sickling and leukocyte adhesion >> vascular tone disruption >> vasoocclusion >> tissue ischemia >> vessel reopening - reperfusion injury >> xanthine dehydrogenase >> converted to xanthine oxidase >> ROS increased tissue damage >> abnormal tissue function >> cellular deaths.

## 4. Heme Oxygenase, Cytoprotection and Therapeutic Potential

Heme oxygenase constitutes a major endogenous countermeasure to oxidative stressors produced by diseases and several other forms of trauma, including toxicant exposure, mechanical, thermal and radiation injury and a plethora of other influences that place oxidative stress on tissue [12,13]. During the past few decades, this enzyme has emerged as a molecule of enormous potential value for the prevention and treatment of a wide range of chronic diseases which historically have been either refractory to currently available therapies or require drugs such as corticosteroids and immunosuppressants with toxic side effects that severely diminish quality-of-life with prolonged use [11]. A major common feature of such pathologies is progressive deterioration of immunoregulatory mechanisms by which inflammatory processes are confined to their roles in normal host defense and are prevented from damaging host tissues, along with increasingly prooxidant tissue environments and symptom severity [12]. Currently available antioxidants that avoid toxic effects are nevertheless mostly ineffective in ameliorating severe, chronic inflammatory conditions since most of these compounds distribute mostly into interstitial spaces of the anatomy, thus diminishing their clinical utility. Representative examples with particular relevance to the present report include chronic inflammatory disorders such as cardiomyopathy, ischemia/reperfusion, endotoxemia, obstructive lung disease, allograft rejection and many others. A unifying common feature of these diseases is an underlying pathology refractory to significant abatement with (for instance) dietary supplements such as vitamin C, the tocopherols (vitamin E) and a diverse repertoire of naturally occurring compounds derived from generally regarded as safe (GRAS) plant sources, such as curcumin, beta carotene and other polyphenols and carotenoids. These are potent dietary antioxidents, but since they distribute to interstitial spaces and are typically overwhelmed by severe inflammatory disease, they are of only marginal value in the treatment of conditions in which regulation of inflammatory processes is severely compromised [11,13,14].

Moreover, although endogenous antioxidants such as glutathione, an intracellular antioxidant active in the cytosol and organelles, counteract oxidative tissue damage to some degree, the protective effect of these compounds is often overwhelmed by reactive oxygen species produced by the host in response to disease and other trauma [15]. The underlying mechanisms by which this occurs are discussed in the present review and prominently include the capacity of HO-1 metabolites of heme, especially bilirubin and carbon monoxide (CO) for reduction in levels of oxidative stress and for counteracting its effects. Particular focus in research and clinical application of HO-1 inducers is currently given to physiological regulation of cardiovascular functions mediated by CO produced by HO-1 and to nitric oxide (NO) expressed by nitric oxide synthetase [15]. Significantly, HO-1 functions both within cells as well as in the circulation and interstitium [13]. The enzyme does not directly degrade or otherwise counteract inflammatory mediators containing reactive oxygen. Instead, the metabolites produced by HO-mediated degradation of heme function as potent regulatory molecules that activate both intra- and extacellular signaling cascades which suppress hyperinflammatory tissue injury—and keep inflammatory processes confined to appropriate biological venues. One of these compounds, bilirubin, is observed in nanomolar concentrations intracellularly, to attenuate the production of reactive oxygen-containing compounds by NADPH oxidase, a primary source of these inflammatory mediators, with outcomes that include dramatic abatement in the occurrence of oxidative tissue injury. For example, studies carried out using both cell culture and in vivo rodent models demonstrate that bilirubin is capable of decreasing oxidative stress on tissues through its inhibitory effect on NADPH oxidase—a major source of potentially pathological reactive oxygen species [16]. A major corollary effect of this inhibition is decreased expression of nitric oxide (NO) synthetase (nos2). Reduced production of NO by this mechanism was further shown to counteract endotoxin (LPS)-mediated hypotension and diminish the occurrence of endotoxic shock and related cardiovascular impairment in animals exposed to LPS [16]. HO-mediated heme processing produces CO at levels which, while subtoxic to host cells and tissues, nevertheless are strongly cytoprotective through its activation of endogenous guanylate cyclase, with downstream conversion of guanosine triphosphate into cyclic guanosine-3-5-monophosphate (cGMP) [17]. cGMP in turn activates signaling cascades that have pleiotropic beneficial effects, particularly in cells of the cardiovascular system, in which the compound acts intracellularly on the p53 protein and Bcl-2 to mediate protection from ischemic damage through the enhancement of cell survival [18]. Taken together, the rapidly evolving research findings and clinical applications of HO suggest that this molecule will emerge into an enormously powerful tool for the long-term management of serious chronic diseases. Nevertheless, such expectations should be tempered with an understanding of the limitations of HO-based therapeutics in the context of technical challenges which remain to be surmounted [19,20,21]. Some of the most prominent limitations, challenges and strategies for working around them are covered below in this article in a section (captioned “Limitations”) focused on how HO-1 induction (particularly with phytochemicals) may suppress inflammatory signal transduction, allowing at least partial restoration of healthy confinement of inflammation to appropriate physiologic settings.

HOs exist in three major functionally distinct isoforms, termed HO-1, HO-2 and HO-3. The best characterized and, at the time of this writing, the enzyme that seems to offer the best chance for development as a preventive and therapeutic tool is HO-1, which is a widely distributed inducible form, considered to have the greatest promise for development as a major tool for preventive medicine and therapy [19].

The value of heme oxygenase (HO) as a “housekeeping” enzyme in the clearance of unreacted heme is emphasized by the fact that heme itself is an effector molecule which regulates the responses of diseased cells and tissues by the activation of molecular oxygen [22,23,24,25,26]. Hence, HOs serve both as tissue detoxifiers and as critical regulators of heme-mediated homeostatic processes.

## 5. Haptoglobin and Hemopexin: Extracellular Countermeasures to Free Hemoglobin and Heme

The cells and tissues of organisms dependent on hemoglobin for oxygen transport have evolved several core countermeasures by which the oxidative stress imposed on living systems as a result of the special biochemistry of the carrier molecules may be controlled. Specifically, adaptive defenses against oxidative stress-mediated vasculopathy by the release of heme, iron and hemoglobin into the blood through hemolysis complements the cytoprotective activity of heme oxygenase and its downstream mediators, CO and bilirubin.

Haptoglobin, an acute phase plasma protein ubiquitously expressed in higher metazoans, provides a major mechanism by which the damaging effects of intravascular or extravascular hemolysis may be reduced in scope and extent. This occurs via a feedback process whereby elevated levels of blood hemoglobin stimulate haptoglobin expression and release into the plasma by cells of several organs, including lung, skin, liver, kidney and adipose tissue, where it binds to free hemoglobin, followed by clearance of haptoglobin–hemoglobin complexes through the splenic reticuloendothelial system [27]. A haptoglobin expression response is a primary adaptive defense against the effects of free hemoglobin released during a hemolytic event. Plasma hemoglobin–haptoglobin complexes are rapidly recognized by the scavenger receptor HbSR-CD163, an inflammatory cytokine-inducible macrophage membrane protein, which directs the complexes into the macrophage/reticuloendothelial endocytotic system for disposal [28]. In addition to the above-described features, endocytosis of hemoglobin via the CD163–haptoglobin pathway acts in an anti-infalammatory role by providing a potent signal for the increased expression of (inducible) heme oxygenase-1 (HO-1), thereby diminishing the tissue burden of oxidative stress, and it additionally contributes to iron metabolism by serving as an inducer of ferritin-1 and provides a major mechanism for iron uptake by macrophages [29].

The protective response against oxidative damage by hemoglobin, as previously described, is complemented by a second line of extracellular defense against free heme: this is an acute phase plasma glycoprotein of hepatic origin, called hemopexin, with strong affinity for the heme molecule [30]. Heme–hemopexin complexes are cleared from circulation and the interstitium through receptor-mediated endocytosis by the CD91 receptor typically expressed by hepatic parenchymal cells, dendritic cells and macrophages [31]. In a manner similar to the biological role of hemoglobin–haptoglobin complexes, heme–hemopexin–CD91 complexes increase HO-1 activity, thus promoting heme degradation, and they additionally promote the activity of anti-inflammatory processes, including iron sequestration by ferritin [32]. The core mechanism by which this cytoprotective process occurs in plasma and cerebrospinal fluid involves the formation of complexes between heme and hemopexin, which binds heme with high affinity, following parrthological release of heme from its normal binding proteins as a consequence of cell damage associated with myolysis, hemolysis, internal hemorrhage and other tissue trauma. Recent studies have shown that the receptor of hemopexin–heme in humans is a low-density lipoprotein receptor-related protein (LRP)/CD91, expressed in a variety of cell types, such as syncytiotrophoblasts, hepatocytes, neurons and macrophages. These receptors are responsive to interaction with hemopexin–heme complexes, following their formation, resulting in internalizartion of the complexes and subsequent lysosomal degradation and clearance of free heme, with resultant protection against its proinflammatory and oxidative toxicogenic effects [32].

In addition to haptoglobin- and hemopexin-mediated clearance of hemoglobin and free heme, respectively, organisms making biological use of the heme molecule have evolved several other extracellular defenses against its toxicity, such as sequestration by ferritin, when present in inappropriate anatomic settings. For example, ferric Fe(III) heme is capable of transference of binding from hemoglobin to albumin, which, despite the relatively low affinity of albumin for Fe(III) and blockade by haptoglobin, may occur at a sufficient level to allow albumin to serve as a reservoir for heme during acute heme overload such as occurs during hemolytic events [33,34]. Novel preventive and therapeutic interventions to counteract failures in heme degradation and overload of heme and/or hemoglobin are likely to emerge from an understanding of the phenomenon of heme iron-mediated oxidation of serum high- and low-density lipoproteins (HDLs and LDLs) which, in the presence of hydrogen peroxide expressed by activated inflammatory leukocytes, form complexes with free heme in a reaction catalyzed by heme iron which are rapidly cleared from circulation [34,35,36]. The occurrence of these processes is particularly significant for the management of vascular pathologies arising as a result of heme-mediated oxidative burden, as is discussed below.

Another pathway which has evolved to inhibit heme toxicity is represented by alpha1-microglobulin (a1Mg), an evolutionarily conserved plasma protein with immunomodulatory properties produced by cells of the liver and normally degraded by cells of the renal proximal tubules. In the blood, a1Mg is typically complexed with IgA, in which configuration it may bind heme, and it possesses a reductase activity capable of reducing methemoglobin and additionally degrading heme at the cytosolic side of erythrocyte membranes [37,38].

## 6. Heme Degradation Products: Impact on Normal Physiologic Function, Disease Risk and Pathogenesis

As described above, heme oxygenases catabolize heme’s porphyrin ring and iron atom within it to carbon monoxide and ferrous, Fe(II) iron, which is then captured by ferritin, an iron-sequestering protein that allows storage of iron in the inner cavity of the molecule as a ferric-oxo species, a form which poses a diminished toxicogenic hazard to the tissues [4]. Processing of heme by HOs also produces biliverdin, a green-colored tetrapyrrolic bile pigment, that is subsequently metabolized by biliverdin reductase, to bilirubin, a yellow open chain tetrapyrrole, which functions as a major physiologic cytoprotective agent through its potent antioxidant properties with systemic levels regulated by conjugation to glucuronic acid and urinary excretion [39,40,41].

## 7. Biliverdin and Bilirubin

Biliverdin, the tetrapyrrolic bile pigment produced as a primary product of heme oxygenase-mediated processing of heme, is rapidly converted to bilirubin by biliverdin reductase (BVR) in normal healthy individuals. In addition to this major catalytic role, BVR exhibits corollary activity which includes modulation of cell signaling mechanisms involving NADPH-dependent utilization of bilirubin to counteract oxidative stress-related tissue damage. An example of the potential capacity of this molecule for use in clinical medicine is provided by mouse model experiments that describe the cooperativity between bilirubin and another major heme degradation product, carbon monoxide, to inhibit the induction of plasminogen activator inhibitor-1 and ROS production, thereby diminishing the levels of these pro-thrombotic molecules, an effect predictive of a protective capacity against stroke in humans [42,43]. A prominent function of bilirubin in the context of these processes is its ability to strongly regulate sources of reactive oxygen species in animal cells and tissues, confining their production and activity to physiologically appropriate settings [44].

Further, in vitro studies have demonstrated that unconjugated bilirubin at low (nanomolar) concentrations also efficiently scavenges peroxyl radicals in homogenous solutions [45,46]; however, at the time of this writing, the clinical relevance of this phenomenon remains uncertain. Stereochemical evaluations of RBC-derived heme turnover (Figure 1) reveal that biliverdin reductase-a (BVRa) is the predominant BVR isoform in adult mammals, while biliverdin reductase-b (BVRb) is the dominant isoform in the fetus [41,47,48]. Moreover, the unconjugated form of bilirubin is normally detected as the first BR pigment to appear in the bile during fetal development [49], showing that heme catabolism differs in utero in comparison with that of adults. This difference in bilirubin metabolism between adult versus gestating mammals may be explained by an observation that unconjugated BR may not freely cross the placenta but must be excreted without previous conjugation with glucuronic acid into the bile [49]. The importance of bilirubin and the biliverdin reductase-a (BVRa) isoform as essential contributors to general physiological antioxidant defense is further underscored by an observation that biliverdin-reductase gene knockout (BVRa^-/-^) mice which were otherwise physically intact and apparently healthy showed increased levels of endogenous oxidative stress, as evidenced by increased plasma cholesterol ester hydroperoxide and elevated oxidation of peroxiredoxin-2 in red blood cells [50]. The aforementioned results indicate that BVRa^-/-^ mice exhibit a higher degree of oxidative stress systemically, implying that bilirubin significantly attenuates the severity of oxidative stressors under experimental conditions. Bilirubin and its unique redox potency for superoxide quenching possesses an indispensable physiological role, despite being substantially less abundant in comparison with other endogenous or exogenous antioxidant agents such as catalase, superoxide dismutase and glutathione [50,51]. Biliverdin and its reduced form, bilirubin, contribute to multiple cellular and molecular mechanisms essential for normal, healthy life and modulate signaling pathways to affect organ function along with susceptibility to, and pathogenesis of, diseases, including disorders of the liver, kidney, heart and brain vasculatures, and immune responses [15,41]. Bilirubin’s role in maintaining healthy organ function and counteracting the effects of disease and other trauma is primarily based on its strong antioxidant properties, along with other features which make it an excellent regulator of the adaptive response to inflammation associated with sepsis and various organ injuries. Moreover, bilirubin possesses immunosuppressive/immunoregulatory properties, as demonstrated by both experimental and clinical studies [52,53]. It is also worth noting that phycocyyanobilin, an algal phytochemical with close structural homology to bilirubin, also exhibits potent antioxident and cytoprotective qualities, with potential for human clinical benefit [54,55,56,57,58].

## 8. Phycocyanobilin: An Algal Bilirubin Analog

Phycocyanobilin is a blue tetrapyrrole chromophore expressed in cryptomonads, glaucophytes, chloroplasts of red algae and cyanobacteria, particularly *Arthrospira maxima*, *A platensis* and *A. fusiformis* (collectively: Spirulina). In its native state, the compound is a component of the algal phycobiloproteins phycocyanin and allophycocyanin. Its close structural homology with bilirubin and negligible biological toxicity have made Spirulina an increasingly valuable dietary supplement for the maintenance of general health and both preventive medicine and therapy [54,55,56,57,58].

## 9. CO, Iron and Ferritin

Carbon monoxide, while highly toxic at high concentrations and inappropriate anatomical settings, is nevertheless benign at levels produced in the catabolism of heme by heme oxygenase. In this mode, it mediates downstream signaling cascades with hugely beneficial effects on the overall health of an organism. CO interacts with a diverse range of cell types to increase the conversion of guanosine triphosphate into cyclic guanosine 3-5-monophosphate (cGMP) through the stimulation of guanylate cyclase. cGMP in turn functions as a second messenger molecule with the capacity to significantly protect cells against ischemic damage following hypoxic events and augment cell survival under stress conditions via effects on the p53 and Bcl-2 proteins [18].

Ferrous iron, Fe(II), produced by HO processing of heme poses a toxicogenic hazard, which is counteracted by complexing of this metal ion to ferritin, a protein which stores the metal as FeIII, and transferrins, metal-binding glycoproteins that transport iron in its FeII state to sites of utilization such as erythroid red blood cell progenitors [59].

## 10. Heme Degradation Deficiency-Associated Tissue Damage and Major Disease Risks

Although heme possesses several normal physiological functions in healthy mammals, high levels of free heme, which occur under various pathological conditions, are very toxic due to the compound’s prooxidant and proinflammatory properties [60,61,62]. Under physiological homeostasic conditions, the toxic hazard posed by heme is diminished by sequestration of the molecule into structures called “heme pockets” of hemoproteins—prominently, hemoglobin [63]. Nevertheless, increased oxidative stress conditions may catalyze the release of heme prosthetic groups, which are capable of causing toxicity—primarily as a result of the capacity of protoporphyrin IX ring-bound iron to act as a Fenton reagent and generate reactive oxygen species [64,65]. These effects create highly reactive lipid peroxides and protein aggregates, resulting in DNA damage and disruption of lipid bilayers in nuclei and mitochondria, with associated ablation of healthy tissue, metabolic derangements and increased cancer risk [9].

Aspects of free heme reactivity described above result in enhanced sensitization for various cell types to proinflammatory agonists, with outcomes that include tissue injury and major symptoms of often fatal disease conditions such as sepsis and malaria [66,67].

## 11. Heme Degradation Deficiency and Vascular Tissue Injury: The Sickling Disease Paradigm

Further consequences of inefficient and/or defective heme degradation are revealed by characterization of its effects on cardiovascular disease—in particular, vascular injury as a result of elevated levels of oxidative stress created by free heme. Outcomes of a comprehensive investigation of this phenomenon by collaborating researchers at University of Debrecen in Hungary and University of Minnesota in the USA were reported in an article by Belcher et al., in a 2010 edition of Antioxidants & Redox Signaling [8]. These outcomes, summarized below, illustrate how deficiencies in normal regulation of heme metabolism cause debilitating and often fatal disruption in cardiovascular homeostasis. One major mechanism by which these derangements occur develops as a consequence of the underlying biochemistry of how the heme molecule is triggered by reaction with oxygen, causing it to interact destructively with host tissue: oxidation of ferrous Fe(II) iron to its ferric Fe(III) form decreases the affinity of the heme protoporphyrin IX molecule for the four hemglobin subunits (a2b2), resulting in the release of free hemin, which, due to its hydrophobic nature, readily becomes intercalated into cell membranes. Subsequently, the heme ring may be cleaved by cellular hydrogen peroxide, releasing iron in an intensely redox-active form capable of catalytic amplification of the rate and quantity at which reactive oxygen-containing compounds are produced. These molecules significantly perturb normal protein expression, amplifying the expression of proinflammatory trasnscription factors such as nuclear factor kappa beta (NF-kB), through oxidation of proteins, protein thiols, DNA and lipids and disruptively activating multiple cell signaling. These heme-derived reactive oxygen species also cause inappropriately high levels of apoptosis, significantly ablating the tissue of normal, healthy cells and contributing to the deterioration of organ function [68]. Major outcomes of these processes which contribute to vascular damage notably include the capacity of heme-derived oxidants to deplete nitric oxide, causing vasoconstriction, oxidize low-density lipoproteins and enhance the infiltration of the vascular endothelium with platelets, red blood cells and leukocytes—all contributing to the progressive impairment of vascular function [8].

An exemplary paradigm for the pathological consequences of oxidative stress induced by failure to clear free heme is offered by sickle cell disease, a devastating and painful disorder resulting from a mutation in the hemoglobin beta chain resulting in symptoms that include ischemia reperfusion injury and multi-organ damage resulting from vasoocclusion. Two rodent models of sickle cell disease in humans, SþS-Antilles and BERK sickle mice, have yielded particularly valuable insight into the mechanisms by which aberrant levels of unreacted hemoglobin promote the pathogenesis of tissue injury by reactive oxygen species [69]. Investigations of these models revealed that the presence of the beta chain mutation corresponded to low plasma haptoglobin and thus lower resistance to abnormal increases in free hemoglobin, since the hemoglobin-scavenging capacity of this protein normally acts as an adaptive countermeasure in hemolytic diseases [70]. The sickling mutation additionally was observed to correspond with methemoglobin, which is a form of the protein generated by the reaction of nitric oxide (NO) with ferrous Fe(II) hemoglobin, thereby depleting NO with resulting exacerbation of vasoconstriction and the recruitment of cells and platelets to the vascular endothelium, which in turn promotes thrombosis [71]. The sickling mutation also is observed to increase the levels of free plasma ferrous Fe(II) hemoglobin, capable of mediating Fenton reactions, contributing to oxidative tissue damage [72]. These processes occur concomitantly with the release of arginase from lysed erythrocytes, which degrades L-argnine, the substrate for nitric oxide synthetase, thereby further NO reducing availability. This depletion is observed to contribute to comorbidities of sickling disease, notably pulmonary hypertension [73]. Figure 2, which is adapted from Belcher et al. [8], illustrates the major features of sickle cell disease as an example of how failure of normal mechanisms for heme degradation triggers severe inflammatory pathological processes. In the case described here, the inflammatory tissue destruction that begins with sickling-induced hemolysis and progresses through stages of increasing oxidative stress results in vasooclusion, ischemia and ultimately organ failure, extending beyond the deterioration of the vasculature to immune derangements in which excessive production of inflammatory cytokines occurs—notably interleukin-6 (IL-6) and tumor necrosis factor-alpha (TNF-a), platelet activating factor (PAF) and effector molecules such as the CD40 ligand [8].

## 12. Augmentation of Heme Degradation Mechanisms and Cardiovascular Health

A major objective in the ongoing investigation of adaptive processes by which healthy cardiovascular function may be protected from hemolytic damage is the development of modulating existing adaptive cytoprotective pathways that have evolved to protect the heart and associated tissues. Currently, these efforts focus on the genetic role and clinical utilization of HOs, particularly HO-1, for the management of heart disease and related comorbidities, especially ischemic events [74,75,76]. The development of clinical strategies for optimal utilization of these processes has become a dynamic and exciting convergence of basic and applied research, representing the core objectives of “bench-to-bedside” initiatives. The particular importance of HO activity in its central role as the primary mechanism by which tissues of the body are protected from the toxicogenic effects of heme is underscored by findings that the risk of human cardiovascular disease is significantly increased by the presence of HO-1 gene promoter polymorphisms that disrupt normal homeostatic regulation and expression of the enzyme [77]. Indeed, isolated hearts from mice modified to overexpress the rat genomic HO-1 transgene exhibited significantly lower incidence of reperfusion-induced ventricular fibrillation and infarct size as compared to hearts taken from mice without this genetic alteration [75]. Further evidence for cardioprotective effects occurring as a result of increasing the level of heme degradation activity has been provided and demonstrated that CO expressed as an HO-1 metabolite in hypertensive rats significantly ameliorated major symptoms of hypertension in the animals as evidenced by increased coronary arterial pressure and blood flow through vasodilation occurring as a result of CO-mediated increases in cyclic nucleotides by cardiac cells (particularly the vascular endothelium) and NO signaling, with accompanying reductions in vascular resistance and improvements in vascular tone. The investigators also demonstrated that these beneficial effects could be reversed by treatment with HO-1 inhibitors [78].

CO-induced increases in bilirubin are observed to inhibit leukocyte adhesion to vessel walls, platelet aggregation and resulting inflammatory tissue damage. Moreover, CO-driven elevation of cGMP has been demonstrated to suppress vascular smooth muscle cell (SMC) proliferation, thereby also inhibiting neointimal blood vessel occlusion [78], a phenomenon potentially leading to novel countermeasures to vascular remodeling and design of drug-eluting stents, along with clinical strategies for the avoidance of stenting, with resulting improvements in quality-of-life. Corollary effects to the biological activities of both CO and bilirubin described above also include control of NADPH oxidase-mediated reactive oxygen species production and stabilization of cationic (Na^+^, K^+^ and Ca^2+^) membrane gradients, essential for the function of both neurological and cardiovascular tissue. Cardioprotective effects occurring as a result of high production of HOs and their downstream effects include general improvements in all cardiac functions, along with extended viability of mouse-to-rat xeenografts and prevention of balloon and IR injury [79].

## 13. HO-1, CO, Arrhythmias and Sudden Cardiac Death

Carbon monoxide is a vasoactive metabolite produced by both intracellular and extracellular degradation of heme by heme oxygense, along with Fe(II) iron and bilirubin. It is a critically important signaling molecule with enormous toxicological and physiological importance to living subjects [80]. Cardiac tissues express both the HO-1 and HO-2 isoforms of this enzyme, which have evolved as essential components of cellular antioxidant defense and adaptive responses to a broad variety of stressors. The inducible isoform, HO-1, is a particularly valuable stress response heart shock protein [77,81,82], with expression and activation increases observed to be particularly significant as an essential modulator of ischemia/reperfusion events [83]. The tissue protective capacity of HO-1 is underscored by observations that isolated myocardium from homozygous knockout mice for the gene encoding the protein for this enzyme (HO-1−/−) exhibited significantly higher cellular damage and incidence of ventricular fibrillation in comparison with myocardium from heterozygous (HO-1+/−) and wild-type mice (HO-1+/+) in studies using isolated mouse hearts mounted in the Langendorf apparatus and subjected to ischemia followed by reperfusion, thus demonstrating the cytoprotective capacity of this major mechanism for clearance of unreacted heme [74,75]. Related investigations have further validated these findings with observations that CO dose-responsively alters intacellular signal transduction pathways within cardiac myocytes through its influence on the regulation of various ion channel functions, especially Na+, K+ and Ca2+, responsible for arrhytmogenesis and sudden cardiac death caused by ventricular fibrillation [84,85].

These studies [84,85] demonstrate the robust protective effects of CO against ischemia/reperfusion-induced arrhythmias in the heart. These reports further show that pharmacological manipulations of HO-1 expression under controlled conditions may have clinical application in antiarrhythmic strategies. Finally, it is of interest also to note that direct application of CO in isolated working rat heart preparations provided significant cardiac protection via a cGMP-mediated signaling mechanism [84,85]. In addition to the contribution of HO-1 in the pathomechanisms underlying the development of arrhythmias in the ischemic/reperfused myocardium, several other enzymes and ion channels may be responsible for these disease mechanisms [84,85]. At the time of this writing, exploration of these processes has demonstrated that, although a full characterization of ischemia-induced arrhythmias remains to be accomplished, ongoing research is providing an increasingly detailed understanding of the underlying genetic factors contributing to reperfusion-induced arrhythmias, since reperfusion-induced arrhythmias arise significantly due to the sharp influx of ROS resulting from rapid re-oxygenation of heart tissue occurring during reperfusion [84,85]. Indeed, given the observed sequence of these events, it is possible that some bioactive proteins are present at the onset of the reperfusion in their inactive forms in the ischemic myocardium, which may be immediately activated upon reperfusion, causing an ionic imbalance in Na^+^, K^+^, Ca^2+^ exchange mechanisms, thereby stimulating the genesis and development of reperfusion-induced arrhythmias. This, however, is speculative and remains to be evaluated in ongoing research into arrhythmogenesis. This work has notably included studies in which animals were administered treatments which affected the pathogenesis of myocardial response to ischemia/reperfusion, including arrhythmic responses. For example, the authors of this report demonstrated that inhibiting calcium flux into cardiomyocytes via ginkgolide-mediated blockade of the platelet activating factor receptor (PAFR) potently potentiated the ability of the macrolide drug FK506 to suppress cellular activation events that contribute to a reduction in Na^+^, K^+^, Ca^2+^ compartmentalization, arrhythmogenesis and reperfusion-associated tissue injury [84,85]. Similar protection against ischemia/reperfusion-related arrhythmogenesis and impaired cardiac function was shown by the authors of this report by pre-treating hypercholesterolemic rabbits with extract of sour cherry seed kernels, which contain a powerful flavonoid inducer of HO-1 [84,85]. These findings demonstrate that pharmacological amplification of heme degradation processes with corollary value as countermeasures to oxidative tissue damage hold enormous future potential in clinical medicine.

## 14. Role of the Kidney in Heme Clearance and Relationship of Renal Tissue to Hemolytic Disease

The following section describes the consequences of a model disorder in which normal mechanisms for the clearance of free hemoglobin and heme are overwhelmed by greatly elevated levels of these molecules due to massive hemolysis: sickle cell disease. Specifically, the release of hemoglobin and free heme from sickled erythrocytes into the plasma overburdens the capacity of regulatory processes such as the formation of hemopexin-CD91 complexes and subsequent increase in heme oxygnase-mediated heme degradation, which is stimulated by these complexes [86], along with heme-induced increases in the haptoglobin gene expression and hemoglobin clearance through its endocytosis in splenic cells by the CD163–haptoglobin pathway [86]. As shown in Figure 2, the most dramatic deleterious effects of this syndrome occur within the vasculature, making failures in normal heme degradation particularly significant in the scope of cardiovascular medicine. It is nevertheless important to emphasize that abnormal systemic overloads of hemoglobin and heme may cause pathologies in a wide range of cells and tissues. A case in point is the relation of renal health to normal heme degradation. The adaptive countermeasures which have evolved to combat severe hemolysis rely on the kidney as a primary site at which excess hemoglobin is degraded and cleared from the system. Hemoglobin uptake occurs substantially due to the expression by brush border epithelial cells of the renal proximal tubules of two endocytic receptors: megalin and cublin, the products of genes which in humans are located on chromosomes 2 and 10, respectively, with megalin (LRP2) serving as the major receptor for hemoglobin clearance and cublin in an adjunct, backup/supporting role in hemoglobinurea cases [86].

As described previously, alpha1-microglobulin (a1Mg) complexed with IgA participates in heme clearance by binding the molecule and recruiting it to cells of the renal proximal tubules for degradation and excretion. Survival of tubular cells in this process is fully dependent on robust HO-1 production under conditions of intense oxidative stress occurring during episodes of heme overload. Indeed, renal tubulointerstitial injury is a hallmark of syndromes in which HO-1 production and activation is deficient [86].

In the context of the foregoing evidence, it is clear that healthy renal function is profoundly impacted by the ability to effectively remove heme as a major toxicogenic threat. It is therefore worth considering how heme oxygenase activity, the major mechanism for heme degradation, affects kidney function. The major features are summarized as follows:

During the life of a typical human, an age-related reduction in the efficacy of renal function generically known as chronic renal failure (CRF) occurs, which involves a progressively more inflammatory tissue environment with comorbidities—especially hypertension and cardiovascular disease, resulting from systemic accumulation of toxicants and waste filtration deficiencies [87]. The underlying pathogenesis of CRF features increased systemic expression of inflammatory cytokines, notably IL-6, TNF-a and interleukin-1 beta (IL1-b), along with other inflammatory humoral factors, which trigger cellular signaling cascades that include pathways mediated by NF-kB and intracellular enzymes of inflammation such as c-Jun-N-terminal kinase [88]. A major result of these reactions is increased insulin resistance by tissues of the body, causing pathologically elevated serum and subsequent increase in molecules containing reactive oxygen. A major consequence includes oxidative damage to the microvasculature of organs, with kidneys and cardiovascular tissue being particularly sensitive [88]. Very encouragingly, clinical use of heme oxygenase induction has shown mounting promise in blocking the progression of the pathological processes described above. The potential use of pharmacological HO inducers in renal medicine is underscored by the extent to which the HO system inhibits inflammatory tissue damage via a number of pathways, including the capacity of renal HO to suppress the transcription of NF-κB, activating protein-1 and c-Jun-N-terminal kinase (JNK), which are strongly proinflammatory transcription factors contributing significantly to renal disease [89]. Amplification of HO activity has also been observed to exert beneficial effects on both systemic and renal cell expression of inflammatory mediators, especially resistin, IL-1β, interleukin (IL-6) and tumor necrosis factor alpha (TNF-α), by stabilizing the expression of these molecules, a process which also inhibits macrophage infiltration into the tissues of the kidneys [89]. Animal studies have demonstrated that treatment with cobalt protoporphyrin increases HO expression in renal tissue to an extent that significant reduction of microalbuminuria is observed in obese rat models, along with the abatement of renal endothelial dysfunction [89]. An expanding repertoire of benign phytochemical HO inducers with clinical potential are being investigated for application in the prevention of human disease. A summary of these is provided above in the present report within the section describing heme oxygenase, cytoprotection and therapeutic potential [89]. Moreover, published reports described how the hormone adiponectin, derived from adipose tissue present in vertebrate plasma, abated the severity of ischemic renal injury in a mouse model through increasing HO-1 expression [90,91]. These results are particularly significant in the context of the future use of HOs in the management of kidney disorders, since increased adiponectin production occurs as an adaptive countermeasure to several major pathologies, especially cardiovascular disease [90].

## 15. Pulmonary Function and Heme-Related Pathologies

The microvasculature of the lung is an anatomical site in which the first critical step in oxygenation of the tissues occurs. Here, relatively high PO_2_, high pH and low PCO_2_ favor the oxidation of Fe(II) (ferrous) iron to its ferric Fe(III) state to form the oxyhemoglobin complex that transports oxygen to tissues where low pH and relatively elevated PCO_2_ favor the dissociation of O_2_ for diffusion into its sites of utilization in oxidative metabolism. When the capacity to efficiently degrade heme or excessive quantities of heme and hemoglobin perfuse lung tissue, as often happens in physical trauma and chronic obstructive pulmonary disorder (COPD), the resulting disruption of tissue homeostasis may cause fibrosis, with concomitant debilitating and often fatal impairment to pulmonary function, a process that is severely exacerbated in the absence of efficient degradation of heme by HO-1 [92]. Interestingly, curcumin, a spice phytochemical widely used in the human diet, has shown promise in the prevention and treatment of lung disease due to the ability of this compound to potentiate the production and activation of HO-1 through the antioxidant transcription factor Nrf2 [93]. Another study published by Pan et al. [89] shows that Nrf2 (Keap1/nuclear factor erythroid 2-related factor 2) in connection with the HO-1 system may be involved in the anti-inflammatory/antiasthmatic effect of edaravone. Adaptive increases in HO-1 are also observed in other lung disorders. Onset of asthma in human patients, for example, correlates with a concomitant elevation in HO-1 expression by cells of the lung, with a consistency sufficiently stable to allow this increase to be used as a clinical indicator of asthma prognosis [94]. These lines of evidence underscore the importance to pulmonary health of physiologic countermeasures to heme overburden and further emphasize the potential value of therapeutic use of HO-1. Here, the underlying pharmacomechanism for antiasthmatic effects is shown to be a consequence of CO-mediated cAMP and cGMP increases in pulmonary smooth muscle cells (SMC) and resulting airway relaxation and lower contractility [95]. Moreover, as intracellular levels of cGMP and cAMP may be increased using phosphodiesterase inhibitors, the use of such compounds together with HO-1-inducers may augment the desired therapeutic effects by counteracting the oxidative stressors imposed on tissue by unreacted heme. Indeed, the authors of the present report have previously demonstrated the efficacy of such an approach using a macrolide inhibitor of phosphodiesterase derived from *Ginkgo biloba* in an animal model of asthma, with outcomes predictive of human clinical application of similar strategies [96]. It has also been recently shown that HO-1 activity and its products are essential for primary airway epithelial cell (pAEC) survival to maintain homeostasis during inflammatory and apoptotic processes in the lung [97].

## 16. Neurological Function and Heme-Related Pathologies

In addition to effects on cardiovascular, kidney and lung function, the outcomes of incomplete or inefficient metabolism of heme have an impact on the brain and peripheral nervous system. The major hallmark of pathologies triggered by the failure of heme degradation processes is intense oxidative stress created principally by the redox properties of the molecule’s bound iron, along with several other structural features of both heme and hemoglobin, as previously described. This property is particularly relevant to neurological disorders, in which oxidative stress is an underlying driver of their pathogenesis [98]. Along with cardiovascular tissue, both the brain (central nervous system/CNS) and peripheral neurons are extremely sensitive to oxidative challenge. The CNS is normally protected from chemical and microbial stressors by the blood–brain barrier; however, influences that are able to surmount this obstacle have potential to inflict enormous damage. The section below summarizes pathologies in which heme-related redox toxicity is a factor.

A particularly delicate structural/functional attribute of neurons making them susceptible to redox damage and ischemia-reperfusion (IR) injury is the manner in which cationic species, especially K^+^, Na^+^ and Ca^2+^, are distributed across the membranes of neuronal cells, which occurs in a manner that polarizes charge across the membranes, allowing propagation of depolarization waves to travel through the system, enabling the performance of the nervous system’s primary biological tasks [99]. This extreme sensitivity makes it imperative for the evolution of specialized countermeasures for certain kinds of injuries, particularly CNS bleeding with hemolysis and other opportunities for exposure of these tissues to the toxicogenic effects of heme. HO-1 is active in the neuronal tissue of normal healthy organs and thus serves as a first line of defense against hemolytic insult [99]. Moreover, ongoing research reveals strong potential for the clinical use of HO-1 as an anti-ischemic agent. For example, rats in which IR-induced CNS stroke was induced using middle cerebral artery occlusion (MCAO) were treated with an HO-1 expression plasmid active in brain cells. Outcomes included significantly reduced infarct size in IR-affected portions of the rat brains along with dramatically decreased pathological expression of TNF-alpha, a result that demonstrates how increasing CNS HO-1 expression may abate the major symptoms of a brain disease state [100]. Additionally, it was emphasized that the neuroprotective mechanisms of HO-1 are closely related to JNK signaling processes during cerebral ischemia/reperfusion [101] and rosiglitazone afforded such protection.

The use of orally delivered phytochemicals has also shown potential for the treatment of IR-associated damage to neuronal tissue. It was observed that *Ginkgo biloba* extract or its components given to animals in a model of ischemia-induced brain injury improved neurologic function and reduced ischemia/reperfusion-induced injury relative to sham-treated control animals in a process whereby the *Ginkgo biloba* treatment significantly increased neuronal HO-1 expression, resulting in neuroprotection in the brain [102,103]. These results have exciting clinical implications in the context of a previous demonstration by the authors of this report that ginkgolides synergistically interact with the macrolide FK506 to augment the cardioprotective effects of this widely used immunosuppressant drug, thus potentially expanding the range of organ donors for transplant surgery, particularly for cardiac transplant patients [104]. Considered together, the observations described above show that significant neurological pathology may result from unreacted heme and that augmentation of major physiologic countermeasures to this kind of toxicity offers significant promise for improved clinical management of brain disease, along with functional derangements of other organ systems in which oxidative stress and resulting hyperinflammation are major components.

## 17. Phytochemical Enhancement of Heme Degradation in Prevention of and Therapy for Inflammatory Pathologies: Evolving Clinical Strategies

Endogenous processes which have evolved to counteract toxic effects, particularly oxidative stress overburden resulting from incomplete clearance of heme from tissues, offer insight into approaches for reducing the risk of onset and the severity of symptoms of a wide range of diseases. Physiological mechanisms for heme degradation are capable of robust restoration of dysregulated inflammatory processes to levels that benefit an organism, rather than causing damage and disease. A major focus of research into clinical adaptation of heme degradative processes is HO activity, particularly the inducible form of this class of enzyme: HO-1. Indeed, by 2002, the enormous promise of HO-1-based preventive medical and treatment strategies was encouraging to the extent that it was labeled a kind of miracle molecule—at the leading edge in a revolution in clinical strategies based on the amplification of existing endogenous defenses rather than intervention with synthetic drugs which were often toxic, costly and effective only for short-term treatment regimens [78]. Nevertheless, by 2019, the technical challenges in the clinical use of HOs had dampened the initial enthusiasm for broad use of these enzymes at the end of the last century. A summary of these challenges and approaches by which they might be surmounted was published that year by members of Lee Otterbein’s lab at Harvard and Brigham & Women’s Hospital in Boston [19].

Nevertheless, in the two decades during which several major laboratories and corporations worldwide were meeting with frustration in the design of practical clinical methods based on HO activities, increasing success was experienced in configuring the oil and flavonoids of *Prunus cerasus* (sour cherry) into low-cost, side-effect-free, preventive measures and treatments for an increasingly expanding range of severe chronic illnesses, particularly inflammatory eye disorders and cardiovascular diseases. The discovery of the therapeutic properties of this seed came about as a result of an adaptation of its medical use dating from medieval times, to experiments demonstrating its capacity to abate inflammatory damage to ischemic/reperfused rat retinas, with an underlying pharmacomechanism found to be dependent on CO release as a metabolite of HO-1 induced by flavonoid components of the seed kernel [105]. This unexpected finding engendered a program to develop the observed effect for use in clinical medicine. A major starting advantage is that sour cherry fruit is an export in several European, American and Asian countries; nevertheless, the seed is discarded as an agricultural by-product. Over thousands of metric tons of the seed annually are produced—and mostly discarded. Accordingly, the authors of the above-mentioned article [105] recognized that this natural medical material (NMM) represented the basis for an extremely promising technology. Subsequent research has borne out this prediction. The investigators demonstrated the capacity of the oil fraction of the seed to completely protect exposed skin against intense ultraviolet light and further showed that the seed kernel contents exhibited no toxicity at dosages over 200 times the therapeutic range [106], and it was also demonstrated that sour cherry flavonoid-mediated effects counteracted the influence of hypercholesterolemia to preserve myocardial function in a rabbit model [107]. Following the publication of these findings, the investigators undertook human in vitro studies using three-color flow cytometry, showing that seed kernel extract suppressed the production of pathological levels of inflammatory cytokines by CD3+ T lymphocytes in a manner predictive of preventive and therapeutic effects in type 2 diabetes [108] and rheumatoid arthritis [109].

A particularly dramatic example of the capacity of sour cherry seed flavonoids to abate the severity of a major disease through the activity of induced HO-1 is a remarkable phase-1 human clinical study in osteoarthritis published in 2014 [109]. For these studies, a topical preparation of the flavonoid fraction dispersed in the seed oil profoundly abated joint pain, along with systemic inflammatory indicators: C-reactive protein and peripheral blood CD3+ subpopulation representation [110].

Ongoing work continues to underscore the powerful cardioprotective effects of HO-1 induction by sour cherry flavonoids [111], and a very exciting human study published and confirmed the earlier evidence for negligible toxicity of this product and further demonstrated that the material elicited no significant increases in peripheral blood expression of alkaline phosphatase—an enzyme with an activation response highly sensitive to potentially toxicogenic influences. This outcome will be extremely valuable in regulatory consideration of the safety profile of products for human consumption worldwide [59]. A particularly intriguing outcome of experiments was the observation that the cardioprotection afforded to rats by beta carotene was significantly diminished at higher doses. Corollary evidence produced by this work strongly suggested that endogenous HO-1 produced in myocardial tissue correlated with this effect. Although insufficient data were produced by these experiments to define a mechanism for this counterintuitive phenomenon, a strong possibility exists that elevated levels of Fe(II) produced by HO-1 activity were at the root of the process, since beta carotene exhibits toxicogenic properties in the presence of Fe(II), which is significantly diminished when the (normally occurring) transferrin activity is augmented by a ferrous iron-specific chelator, such as tamarind extract. Use of such an extract offers a strong possibility that beta carotene might mediate fully dose-responsive cardioprotection, in which the full benefits of endogenous HO-1 activity might be utilized with no interference from Fe(II). This is speculative at the moment but grounded in solid evidence [14,112]. These results also raise a cautionary note with respect to the role of iron metabolism in the modulation of HO activities.

Previous studies to define the relationship between systemic levels of Fe(II) and HO activity and revealed that influences stimulating the induction of HO-1, if prolonged for extended time periods, caused significant decreases in transferrin receptor (TR) expression, with resulting upregulation of Fe(II) levels, using a model reasonably predictive of clinical outcomes [113]. This core process thus has the potential to greatly diminish the clinical efficacy of HO-based treatments, unless the above-described iron-dependent effect is compensated for. Additionally, *Prunus cerasus* and/or *avium* extract and its components are an alternative medicine used traditionally for the amelioration of oxidative stress in neuropathy and various inflammatory processes including gastrointestinal and cardiovascular systems. The components of *Prunus cerasus* and/or *Prunus avium* are very active phytochemicals utilized in “in vitro and in vivo” biological models to explore their possible mechanisms of action in various tissues and diseases [114]. Studies show that a reduction in the proinflammatory TNF-alpha and IL-6 levels and an elevation in the anti-inflammatory factor of IL-10 and oxidative stress amelioration can be important mechanisms responsible for the anti-inflammatory potential of the components of *Prunus cerasus*. *Prunus cerasus*’ components, separately or together, may significantly reduce inflammatory processes and pain thresholds in various tissues [114,115].

## 18. Ginkgo Biloba, “Ginkgoflavonoids” and HO-1

*Ginkgo biloba*, nowadays the widely cultivated Chinese species, has an evolutionary lineage which dates back to the early Jurassic era, around 180 to 190 million years ago, and over this period of time, the tree has undergone several changes, leading many scientists to refer the ginkgo as a “living fossil”. Currently, *Ginkgo biloba* is an adaptive species and grows well in most parts of the world, including in the Mediterranean climate. The extract of *Ginkgo biloba* and/or its ginkgolide components, e.g., EGb761, BN52021, BN5739 and BN52063, possess many beneficial effects on the function of several injured and diseased cells and organs, including the nervous [116,117,118,119,120], renal [121,122,123] and cardiovascular [124,125,126,127,128,129,130] systems.

Many potential mechanisms in various tissues have been proposed to explain the protective actions of ginkgolides, including their effects on the platelet activating factor [131,132,133], generation of reactive oxygen species [134,135] and different signal transduction processes [135,136,137,138,139]—one of these actions has been recently connected to the role of the HO-1 system [138,140]. Thus, the effects and action mechanisms of *Ginkgo biloba* extract (EGb-761) were studied in ischemia/reperfusion-induced injury in rats [138], and it was found that levels of of creatine kinase-MB, lactate dehydrogenase, troponin T, TNF-α, IL-6 and IL-1β were significantly reduced in the drug-treated group in comparison with the drug-free control values. Additionally, expression of caspase-3 and Bax in the EGb 761-treated group was at lower levels than those of the drug-free control group, whereas the expression of Bcl-2, p-Akt and HO-1 and nuclear protein Nrf2 was increased in the untreated control group. These results show that EGb 761 inhibits apoptotic cell death and protects the ischemic/reperfused myocardium via the activation of the Akt/Nrf2 pathway, increasing the expression of HO-1, leading to a reduction in oxidative stress and repressing inflammatory signal transduction mechanisms.

## 19. Limitations

As described above, HO-1 inducers offer an enormously valuable approach to altering the redox environment within cells and the interstitium of biological systems in ways that potently inhibit tissue damage by ROS and other chemical species bearing unpaired electrons. A major cautionary note in the application of such methods—particularly for the “bench-to-bedside” transition of research outcomes to clinical practice—is that this class of reactive compounds is essential for the broad classification of signal transduction known as redox signaling, in which levels of particular reactive compounds determine major metabolic outcomes or, in the case of primary immune responses, may function as antimicrobial agents. For example, nitric oxide (NO) is ubiquitously required for a wide range of healthy functions, and agents that may quench its reactivity may have enormously adverse health effects, particularly for cardiovascular tissue. Moreover, as described above, Fe(II) produced by HO processing of heme poses a toxicogenic hazard and, additionally, Fe(II) produced by HO-1 activity has actually been observed to diminish the cardioprotective activity of beta carotene [112]. These effects impose a built-in limitation to the use of antioxidant compounds and the modulation of redox signaling.

## 20. Future Directions

The enormous promise of a novel, wide-ranging preventive and therapeutic technology based on the phytochemical augmentation of HO-1 activity represents a potential revolution in biomedical science. Indeed, the optimistic predictions by Morse and Choi in 2002 [78] that this revolution was at hand remain valid and constitute compelling rationale for funding bodies to invest in its goals. This spirit must nevertheless be tempered by the analyses by Leo Otterbein and colleagues in 2018, warning that the challenges to the full optimization of this technology remain formidable [11].

## Figures and Tables

**Figure 1 ijms-21-09698-f001:**
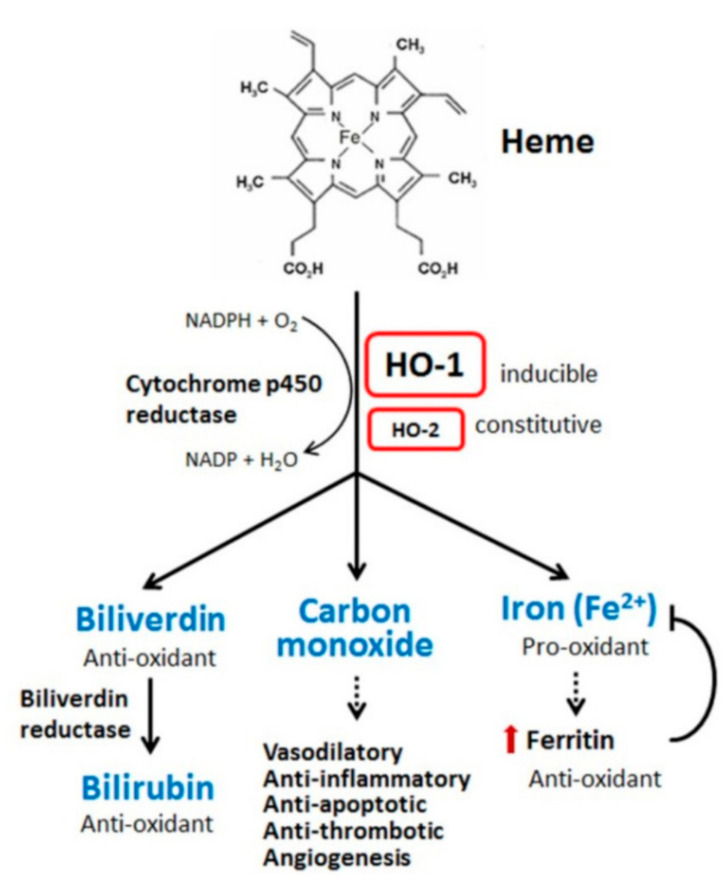
Heme oxygenase (HO-1)-mediated degradation of heme. HO-1 degrades heme into biliverdin, which is subsequently metabolized by biliverdin reductase into bilirubin, a potent physiological antioxidant. The processing of heme by HO-1 additionally releases Fe(II) ferrous iron, which is sequestered by ferritin, a protein that reduces Fe(II) toxicity, and CO, which mediates vasodilatory, anti-inflammatory, anti-apoptotic, anti-thrombotic and pro-angiogenic activities, which collectively constitute a major cytoprotective adaptive response to both external and endogenous stressors. Figure is adapted from Shih-Kai Chiang et al. [7].

**Figure 2 ijms-21-09698-f002:**
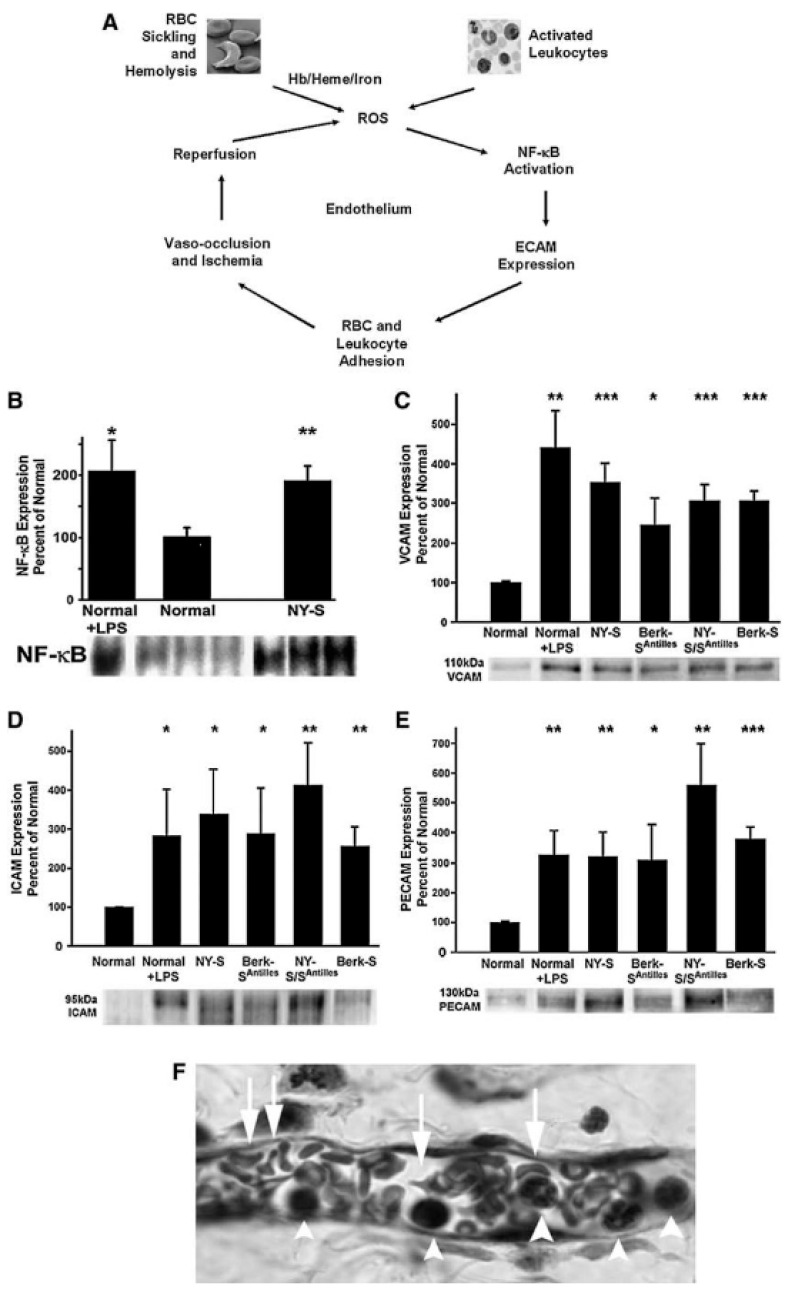
Hemolysis, oxidative stress, inflammation and adhesion lead to vasoocclusion and ischemia/reperfusion injury in sickle cell disease. (**A**) A vicious cycle of oxidative stress, inflammation and vasoocclusion in sickle cell disease is initiated by hemolysis of sickle RBCs, releasing hemoglobin and heme into plasma, iron from which catalyzes additional ROS production. Free heme further reacts with activated leukocytes to amplify the expression of ROS and proinflammatory cytokines, which promote endothelium-derived ROS, activating endothelial NF-kB-dependent pathways and endothelial cell adhesion molecule (ECAM), vascular cell adhesion molecule-1 (VCAM-1), intracellular cell adhesion molecule-1 (ICAM-1), P-selectin and others, promoting vasoocclusion and subsequent tissue ischemia. Thus, these vessels can subsequently reopen, and reperfusion leads to the conversion of xanthine dehydrogenase to xanthine oxidase, promoting extra ROS production. (**B**) Electrophoretic mobility shift assay (EMSA) demonstrates that LPS injection induces NF-kB in the lungs of transgenic New York sickle (NY-S) mice. This figure shows the adhesion molecule bands from one representative lung from each model and a summary bar graph. The bar graph plots the mean + SD adhesion molecule expression for each mouse model as a percentage of normal control mice (**B**–**E**). (**F**) Histology of venule in the dorsal skin of transgenic sickle mice after 1 h of hypoxia and 1 h of reoxygenation. The figure also shows a venule with a suspected vascular obstruction. White arrowheads, leukocytes that appear to be adherent to the vascular endothelium; white arrows, misshapen RBCs inside the venule. * *p* < 0.05; ** *p* < 0.01; and *** *p* < 0.001. The events shown in this figure and described above are summarized in the schematic sequence of major events diagram shown above in this paper. This scheme shows the major sequential processes underlying the pathogenesis of sickling disease, as described here in Figure 2. Figure is adapted from Belcher et al. [8] with the permission of the publishers (Mary Ann Liebert, Inc.).

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
