# Peer review of "Heme Degradation in Pathophysiology of and Countermeasures to Inflammation-Associated Disease"

_ijms, 2020, doi:10.3390/ijms21249698_

Round 1

Reviewer 1 Report

This is a quite long review that would greatly benefit from being accompanied by schemes that summarize the various processes and from a greater attention to the rigor of definitions.

Overall, the text does not make a clear distinction between extracellular and intracellular processes and proteins. It is also unclear the choice of organisms selected as examples (from humans and their diseases to ginkgo biloba).

In its present version, the manuscript does not give the impression of a very authoritative contribution.

A non-exhaustive list of issues to be addressed in a revised version is provided below.

Abstract: Ferrous, reduced iron, FeII (or, better Fe(II)), Fe2+: all these terms are synonymous. Just choose one of them.

Section 1: Overall it sounds naïve with several not rigorous claims (in particular at lines 44-58). In addition, the following points deserve attention.

Line 37: the porphyrin ring of hemes is not planar

Line 39: wrong definition of metalloporphyrin, which is a broader class of molecules

Section 2: arbitrary selection of hemeproteins to be listed. Why not myoglobin, for example?

Legend to Figure 1. Another example of not rigorous description: “ … ferrous iron (Fe2+), which complexes with ferritin, an iron-sequestering protein complex that reduces Fe2+ toxicity” 

Lines 183-185: Even worse: “…  ferrous (Fe2+ ) iron, which is then complexed with ferritin, a carrier protein  that allows storage of Fe2+ without tissue toxicity.” What is stored inside ferritin is a ferric-oxo species caged inside the inner cavity. The situation is very different from Fe2+ complexation.

Section 8: “… of this metal ion to ferritin, a glycoprotein which stores the metal …” Only plasma ferritin is glycosylated. All other ferritins (cytoplasmic and mitochondrial ferritins are not glycosylated).

Line 161:  including iron sequestration buy ferritin ->   including iron  sequestration by ferritin

Author Response

Responses (Reviewer Number 1.) have been attached.

Reviewer 2 Report

Major concerns

In general, the manuscript reviewed numerous studies and provided some informative summaries and suggestions. In the review titled as “Heme degradation in pathophysiology of and countermeasures to inflammation-associated disease”, except the renal parts, the author didn’t have clear descriptions to address pathophysiology due to inflammation, despite that the authors provided the role of heme degradation/HO-1 in the inflammation-associated disease. Here are some specific comments.    

  1. Line 92-95, “Currently available antioxidants that ….thus diminishing their clinical utility”. Some examples of cases or diseases and releated antioxidants or compounds should be presented.
  2. Line 102-106, “One of these .. in occurrence of oxidative tissue injury.”. some specific cases should be noted.
  3. Line 140, “stimulates haptoglobin expression into the plasma by cells…”, this description is not appropriate and is suggested as “expression and release into ..”
  4. Line 159-161, “heme-hemopexin-CD91 complexes increase HO-1 activity…”. Please specify the mechanisms if possible.
  5. Line 161, by ferritin”.
  6. Line 195, “counteract oxidative stress-related tissue damage”, Please specify the damages or diseases, if possible.
  7. Line 229-235, the whole paragraph is impudent, looks independent and irrelevant to the core of the manuscript.   
  8. Line 269, were reported in an ..
  9. In Figure 2, the legend is too long. Some of the descriptions can be moved in to the content.
  10. Line 386, Scientific evidences suggested…
  11. Line 395, signal mediated mechanisms
  12. Line 396-411. This whole paragraph is too general and vague. Please specify the genes/proteins.
  13. Line 447-448, “Very encouragingly, clinical use of heme oxygenase induction has shown mounting promise in blocking progression of the pathological processes described above.” Please present some cases to specify the chemicals of HO induction and renal damages.

Author Response

Responses to Reviewer Number2. Please, see the attachment.

Round 2

Reviewer 1 Report

The authors have introduced a minimal set of changes to partially account for the reviewers criticisms.

Unfortunately, I can access a new submission without figures, and therefore I cannot establish whether ther was an improvement of the graphical presentation. For sure, only 2 figures in a 30-page long review are not many.

As far as the text is concerned please note that at line 67: the word cytochromes is repeated twice.

Author Response

To Reviewer Number 1:
